# DISCOVERING INTERPRETABLE BIOLOGICAL CONCEPTS IN SINGLE-CELL RNA-SEQ FOUNDATION MODELS

## ABSTRACT

Single-cell RNA-seq foundation models achieve strong performance on downstream tasks but remain black boxes, limiting their utility for biological discovery. Recent work has shown that sparse dictionary learning can extract concepts from deep learning models, with promising applications in biomedical imaging and protein models. However, interpreting biological concepts remains challenging, as biological sequences are not inherently human-interpretable. We introduce a novel concept-based interpretability framework for single-cell RNA-seq models with a focus on concept interpretation and evaluation. We propose an attribution method with counterfactual perturbations that identifies genes that influence concept activation, moving beyond correlational approaches like differential expression analysis. We then provide two complementary interpretation approaches: an expert-driven analysis facilitated by an interactive interface and an ontology-driven method with attribution-based biological pathway enrichment. Applying our framework to two well-known single-cell RNA-seq models from the literature, we interpret concepts extracted by Top-K Sparse Auto-Encoders trained on two immune cell datasets. With a domain expert in immunology, we show that concepts improve interpretability compared to individual neurons while preserving the richness and informativeness of the latent representations. This work provides a principled framework for interpreting what biological knowledge foundation models have encoded, paving the way for their use for hypothesis generation and discovery.

## 1 INTRODUCTION

With the development of high-throughput genomic technologies, the availability of large-scale biological datasets has exploded (Barrett et al., 2005; Regev et al., 2017). Among the available modalities, single-cell RNA sequencing (scRNA-seq) captures information about gene expression within individual cells, providing detailed insights into underlying biological functions and improving our understanding of cells, diseases, and drug action mechanisms (Jovic et al., 2022; Wang et al., 2023).

Deep learning models trained on these large scRNA-seq datasets have demonstrated their potential in key tasks such as perturbation response prediction (Cui et al., 2024) and multi-batch integration (Lopez et al., 2018). While some models are trained with additional constraints on interpretability (Ruiz-Arenas et al., 2024; Bourgeais et al., 2022; Zhang et al., 2023), several widely used models are black-boxes (Cui et al., 2024; Lopez et al., 2018) and require post-hoc approaches to understand their predictions. Post-hoc explainability methods for scRNA-seq models are limited, which impacts the practical utility of black-box models. More tools could help uncover internal decision-making processes, allowing biological insight, discovery of knowledge, and in silico hypothesis testing (Conard et al., 2023).

Sparse dictionary learning has recently emerged as a promising approach for extracting interpretable concepts from the latent spaces of deep learning models. Initially introduced in the context of language models (Huben et al., 2023) and vision models (Fel et al., 2023), this technique has been rapidly extended to biological models (Adams et al., 2025; Schuster, 2024). A major challenge in

---

Preprint, under review.

applying this approach to biology lies in interpreting the learned concepts. Unlike in textual or visual domains, where concepts often have intuitive semantic meaning, biological sequences are not inherently human-interpretable. To address this, Adams et al. (2025) proposed an interface for visualizing concepts in proteins to support interpretation. Compared to protein sequences, scRNA-seq is usually treated as an unordered sequence of genes and is less convenient to visualize. Schuster (2024) uses automatic pathway enrichment to map scRNA-seq concepts to known biological pathways. However, we argue that pathway enrichment restricts scRNA-seq concepts to prior structured knowledge, potentially overlooking other biologically meaningful signals, such as specific cell types or even novel biological insights.

In this work, we investigate the use of Sparse Autoencoders (SAEs) to extract interpretable concepts from cell embeddings in scRNA-seq models. Using two large immune cell datasets (Cross-tissue Immune Cell Atlas (Domínguez Conde et al., 2022) and Tabula Sapiens Immune (Consortium* et al., 2022)) alongside two state-of-the-art models with distinct architectures (scGPT (Cui et al., 2024) and scVI (Lopez et al., 2018)), we explore the **interpretability**, **stability**, and **usefulness** of the extracted concepts, critical characteristics for practical utility. Interpretability refers to whether the concepts capture meaningful biological patterns such as tissues (e.g., "Colon"), cell types (e.g., "Neutrophil"), biological processes (e.g., "Cytosine biosynthetic process"), or other molecular signals that can be interpreted by domain experts. Stability evaluates whether similar concepts are consistently recovered when SAEs are trained on different datasets. Usefulness assesses whether the concepts preserve biological signal compared to original neuron activations and whether they support interpretable downstream analyses.

We introduce novel tools for interpreting concepts in scRNA-seq models, bridging the gap between computational biology and explainable AI. First, we propose an **attribution-based method with counterfactual perturbations** to identify genes that differentiate cells activating the concept from similar cells that do not. Our approach goes beyond Differential Gene Expression Analysis (DGEA) proposed in Schuster (2024) and helps to distinguish genes that influence concept activation from spurious correlations. Building on attribution results, we propose **attribution-based Gene Set Enrichment Analysis** (GSEA), which uses the GSEA algorithm (Subramanian et al., 2005) with attribution scores instead of traditional fold-change scores. Unlike fold-change scores, which emphasize genes with large expression differences, attribution scores highlight genes that are most influential from the model's perspective, enabling more meaningful pathway prioritization. To go beyond pathway enrichment, we developed and deployed a **web-based visualization tool** to facilitate expert interpretation and conducted an interpretation study in collaboration with an immunology expert.

Our findings demonstrate that concepts from SAEs are more interpretable than individual neurons from the model and align well with biological signals such as cell types and biological processes. In addition, we identify a set of stable concepts across datasets. Finally, we find that the resulting concept space preserves predictive performance in cell type and cell cycle phases and allows for more interpretable classification. Our work demonstrates that SAEs offer a promising approach for uncovering biological signals encoded in scRNA-seq deep models. Their interpretable latent representations may, in future work, support the generation of novel biological insights.

## 2 BACKGROUND ON CONCEPT EXTRACTION

Concept extraction method is illustrated in Figure 1.1. We consider a deep learning model $f : \mathbb{X} \rightarrow \mathbb{A}$, that maps inputs from $\mathbb{X}$ to a representation space $\mathbb{A} \subseteq \mathbb{R}^d$ of dimension $d$. In our case, the input space is the gene expression space with $G = g_1, ..., g_m$ the gene symbols and $x \in \mathbb{R}^m_+$ their corresponding expression level. The representations of $n$ samples form a matrix $A \in \mathbb{R}^{n \times d}$. Concept extraction is framed as a dictionary learning problem where $A \in \mathbb{R}^{n \times d}$ is approximated using a decomposition method $(U^*, D^*) = \arg\min_{U,D} ||A - UD^T||_F^2$, with additional constraints on $U$ or $D$ that promote interpretability. The objective is to learn a dictionary $D \in \mathbb{R}^{d \times c}$ of $c$ concepts such that the activations can be reconstructed as sparse linear combinations of concepts in $D$, with $U \in \mathbb{R}^{n \times c}$ the corresponding coefficients.

Several decomposition approaches can be used, such as non-negative matrix factorization (NMF), independent component analysis (ICA), and sparse auto-encoders (SAE). Following the results in Fel et al. (2025b), we use SAEs, which achieve better reconstructions at a fixed sparsity level. We further choose to use TopK SAE following the work of Gao et al. (2024), which simplifies tuning

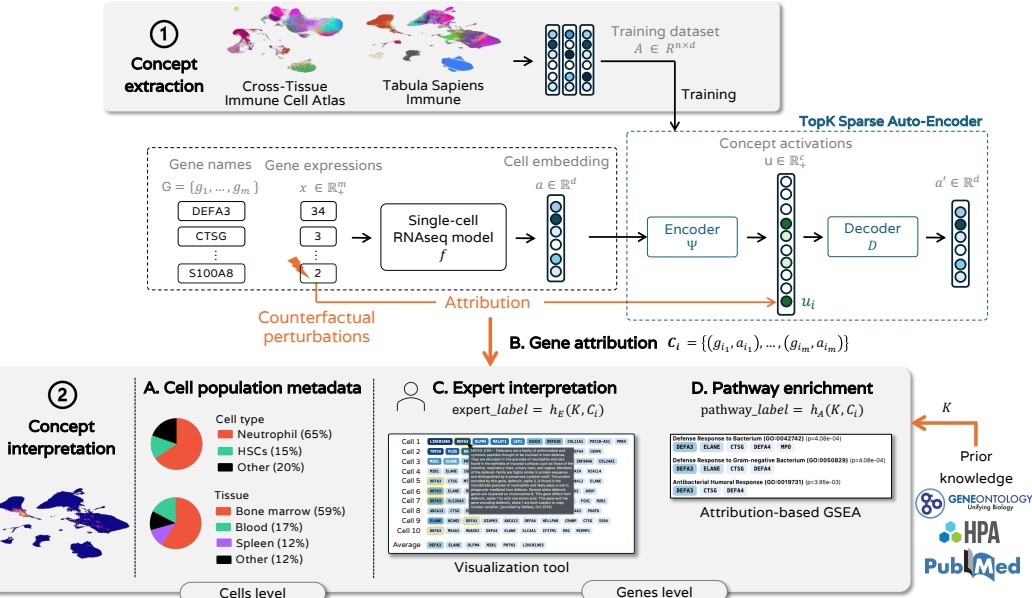

Figure 1: **Illustration of the methodology to extract and interpret biological concepts from scRNA-seq models.** (1) Concepts are extracted by training Topk SAEs on two scRNA-seq datasets. (2) We introduce a set of methods to biologically interpret concepts. (A) Characteristics of the cell population that activate a concept based on available metadata per cell. (B) Attribution method based on counterfactual perturbations to score genes according to their importance for concept activation. (C) Expert interpretation of the concept based on the gene attribution results and prior knowledge. We developed and deployed a visualization tool to facilitate manual interpretation. (D) Attribution-based pathway enrichment detects pathways enriched with genes that influence concept activation.

and improves the reconstruction-sparsity frontier compared to vanilla SAE. Experiments with semi-NMF are given in Appendix B.

Sparse auto-encoders (SAEs) first map $A$ to $U$ with $U = \Psi(A) = \sigma(AW + b)$, where $\sigma(\cdot)$ is a non-linear function, and reconstruct $A$ with $A' = UD^T$. Topk SAEs enforce the sparsity of $U$ by selecting $k$ concepts per sample with $\sigma(x) = ReLU(Topk(x))$.

## 3 METHODOLOGY TO INTERPRET A CONCEPT

In this section, we present our methodology for concept interpretation. We begin by characterizing concepts at the cell population level using available metadata per cell (Section 3.1, Figure 1.2.A). To identify the genes driving concept activation, we propose an attribution approach based on counterfactual perturbations (Section 3.2, Figure 1.2.B). The resulting gene set is then interpreted either by domain experts (Section 3.3, Figure 1.2.C) or algorithmically via prior biological knowledge; in this work, we propose attribution-based pathway enrichment (Section 3.4, Figure 1.2.D).

### 3.1 CELL-LEVEL OVERVIEW WITH METADATA

We propose a first approach to characterize a concept given the cell population that activates the concept. Some metadata are typically available at the cell level, such as the tissue and patient of origin, as well as the annotated cell type. Given the $j$ cells that activate the concepts and their $l$ one-hot metadata labels $M \in \{0,1\}^{j \times l}$, we compute for each metadata the ratio of cells with a positive label $r = \frac{1}{j} \sum_{i=1}^{j} (M_i^T)$. High ratios highlight the concept specificity for the corresponding metadata. While metadata enrichment offers a preliminary means of rapidly analyzing cell populations, concepts are intended to convey more granular biological meaning, such as biological processes,

which are defined through gene-level activity. Moreover, metadata enrichment is limited to prior knowledge and does not support biological discovery.

## 3.2 GENE-LEVEL UNDERSTANDING WITH ATTRIBUTION

To enable precise concept interpretation, we aim to identify the genes that specifically drive concept activation. We propose to leverage attribution methods with counterfactual perturbations. This approach extends beyond Differential Gene Expression Analysis (DGEA) proposed in Schuster (2024) and helps to distinguish genes that influence concept activation from spurious correlations.

For a concept $i$, a cell $x^p$, and a baseline cell $x^c$, the attribution method explains the concept activation score given by $(\psi \circ f)_i$ for $x^p$ by computing one score per gene : $a((\psi \circ f)_i, x^p, x^c) \in \mathbb{R}^{|\mathbb{X}|}$. The higher the score, the more important the gene.

As stated by Mamalakis et al. (2023), attribution methods are highly sensitive to the baseline $x^c$, which should be carefully chosen depending on what we aim to explain. We propose using counterfactual baselines to detect the signal that distinguishes cells that activate a concept from similar cells that do not activate the concept (*counterfactual*). We define a counterfactual of the cell $x^p$ for concept $i$ as the closest cell that does not activate the concept (Equation 1).

$$x^c = \arg \min_{x^j} \left\{ ||f(x^p) - f(x^j)||_2 \, \middle| \, (\psi \circ f)_i(x^j) = 0 \right\} \tag{1}$$

Following Occlusion method (Zeiler & Fergus, 2014), for a concept $i$, a prototype cell $x^p$, and a counterfactual cell $x^c$, we perturb each gene one by one, replacing the expression $x_l^p$ of gene $l$ with the expression in the counterfactual cell $x_l^c$ and compute the variation in concept activation. The equation of the attribution score for gene $l$ is given in Equation 2.

$$a_l((\psi \circ f)_i, x^p, x^c) = (\psi \circ f)_i(x^p) - (\psi \circ f)_i(\tilde{x}^p) \text{ with } \tilde{x}_j^p = x_l^c \text{ if } j = l, \text{ otherwise } x_j^p \tag{2}$$

For more robustness, instead of taking a single counterfactual per cell, we select the $N_c$ closest cells and average attribution scores. For each concept, we compute attribution scores for $N_p$ prototype cells and average, which gives $C_i$, the list of $m$ genes sorted by attribution scores (Equation 3).

$$C_i = \{(g_{l_1}, a_{l_1}), \cdots, (g_{l_m}, a_{l_m})\} \text{ with } a_{l_1} \geq a_{l_2} \geq \cdots \geq a_{l_m} \tag{3}$$

## 3.3 EXPERT INTERPRETATION WITH INTERACTIVE VISUALIZATIONS

We first rely on domain expertise to interpret the set of genes. Specifically, for each concept $i$, a biologist $f_E$ uses their own knowledge, external resources $K$, and results of gene attribution $C_i$, producing an expert label : $expert\_label = f_E(K_E, C_i)$. To support this process, we developed an interactive interface to visualize the most relevant genes given $C_i$. External knowledge $K$ is partially integrated by displaying gene description from the NCBI Gene database (Brown et al., 2015) upon hovering over each gene. Additional knowledge sources will be incorporated in the future to further assist experts in concept interpretation.

## 3.4 ATTRIBUTION-BASED PATHWAY ENRICHMENT

In addition to expert interpretation, algorithms can be used to assign labels to concepts by leveraging prior knowledge in an automated way. Formally, for a given concept $i$, an algorithm $f_A$ integrates prior knowledge $K$ and gene attribution results $C_i$ to output a label: $algo\_label = f_A(K, C_i)$. In this work, we use a widely used algorithm in computational biology, Gene Set Enrichment Analysis (GSEA, Subramanian et al. (2005)). GSEA operates on a ranked list of genes and evaluates, for each biological pathway from a given ontology, whether genes associated with the pathway are clustered together at the top of the list. Genes are usually ranked by the fold-change value obtained with a Differential Gene Expression Analysis. Instead, we propose ranking genes by their attribution values to prioritize pathways that include the most influential genes for the concept. We refer to this method as attribution-based GSEA. We used the *prerank* method from the GSEApy package (Fang

et al., 2023) to implement $f_A$, the Biological Processes from the Gene Ontology (Ashburner et al., 2000) as prior knowledge $K$ [1], and attribution scores $C_i$ as described in Section 3.2.

# 4 EXPERIMENTS

## 4.1 DATASETS AND MODELS

We studied two generative models for scRNA-seq data with different architectures. **scVI** (Lopez et al., 2018) is a variational auto-encoder; we used the checkpoint[2] provided by CellxGene Census Program et al. (2025), which encodes sequences of 8 000 genes. The second model, scGPT (Cui et al., 2024), is a Transformer-based model that encodes sequences of 1 200 genes. We used the "whole-human" checkpoint from the official repository [3] and explored the last cell embedding token, corresponding to the special token "CLS".

The study focused on immune cells, with the **Tabula Sapiens Immune** dataset (Consortium* et al., 2022) containing 592 317 cells and the **Cross-tissue Immune Cell Atlas** (Domínguez Conde et al., 2022) containing 329 762 cells. Both datasets are annotated with cell type, tissue, and patient. Additional descriptions and visualizations for both models and datasets are given in Appendix A.

## 4.2 TRAINING TOPK SAEs AT DIFFERENT SCALES

We explored several expansion factors with a fixed sparsity level, which we found to minimize the number of dead concepts at the end of training. The latent dimension of scGPT is $d = 512$, we trained 4 Topk SAEs with $c = 1000, 2000, 5000$ and $10000$, with $k = 16, 32, 80$ and $160$ respectively. These values are in line with common practices in the literature, with expansion factors ranging approximately from 2 to 20. The latent dimension of scVI is $d = 50$, we trained 5 Topk SAEs with $c = 200, 500, 1000, 2000$ and $5000$, with $k = 3, 8, 16, 32$, and $80$ respectively. SAEs are trained on each dataset separately. The results for SAEs trained on the Tabula Sapiens Immune dataset are displayed in Figure 2, hyperparameters and metrics for both models and datasets are given in Appendix B.

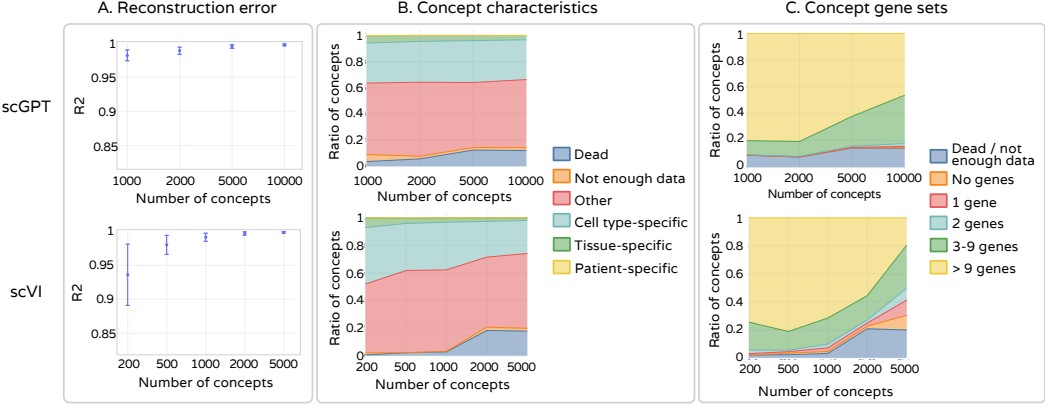

Figure 2: **Evaluation of Topk SAEs trained at different scales**. Results for SAEs trained with the Tabula Sapiens Immune dataset, for scGPT (top) and scVI (bottom). (A) Cell embedding reconstruction quality as measured by the $R^2$ score. (B) Concepts characteristics at the cell level based on metadata. (C) Gene set characteristics based on attribution results ($attribution > 0.05$). *Not enough data* means that less than 100 cells activate the concept (We do not expect a biological signal to appear in such a small portion of the dataset).

---

[1] GO_Biological_Process_2025

[2] s3://cellxgene-contrib-public/models/scvi/2025-01-30/homo_sapiens/model.pt

[3] https://github.com/bowang-lab/scGPT/blob/main/README.md

**Reconstruction error.** We evaluated the error of cell embedding reconstruction with the $R^2$ score. A score of 1 indicates perfect reconstruction, while a score of 0 indicates a reconstruction no better than the mean cell embedding. All SAEs achieved a nearly perfect reconstruction with $R^2$ greater than 0.95 (Figure 2.A). As expected, reconstruction quality improved with the number of concepts.

**Concepts characteristics.** As introduced in Section 3.1, we used cell metadata to characterize concepts at the cell population level (Figure 2.B). For example, a concept is labeled as "tissue-specific" if at least 70% of the cells activating the concept come from the same tissue. For both scVI and scGPT, a large proportion of concepts are specific to a cell type, which is expected, as cell type is a strong signal in gene expression. Gene-level interpretation is necessary to further characterize the concepts.

**Gene set characteristics.** Following the methodology introduced in Section 3.2, we computed gene attribution scores for each concept to obtain $C_i$ (Equation 3). We averaged scores over 10 cells with the highest concept activation and $N_C = 3$ and filtered out genes having little impact on concept activation (attribution lower than 0.05). For most concepts, the attribution method detects more than 3 genes having an effect on concept activation (Figure 2.C). As a comparison point, biological processes of the Gene Ontology are linked on average to 3.6 genes and 9.1 genes when considering, respectively, the 1200 genes of scGPT and 8000 genes of scVI. We further compared the gene sets obtained with attribution to gene sets obtained with Differential Gene Expression. Deletion curves confirm that attribution more precisely identifies the genes that influence concept activation, allowing us to focus on the most relevant genes (details curve in Appendix C).

This initial analysis shows that concepts are often specific to cell types, but also capture other signals that require more subtle investigation. The gene sets obtained with attribution contain enough genes to enable further interpretation. We observe a limitation in the expansion factor for scVI, with an increase in dead concepts and fewer important genes per concept for $c = 2000$ and $c = 5000$. Therefore, we used $c = 5000$ ($k = 80$) for scGPT and $c = 500$ ($k = 8$) for scVI in the rest of the study, which corresponds to expansion factors of approximately 10, consistent with prior work.

## 4.3 CONCEPTS ARE MORE INTERPRETABLE THAN NEURONS

In this section, we use the methods introduced in Section 3 to evaluate the interpretability of concepts from SAEs and compare them with individual neurons. We used SAEs trained on the Tabula Sapiens Immune dataset, with $c = 5000$ ($k = 80$) for scGPT and $c = 500$ ($k = 8$) for scVI.

**Expert interpretation study.** We conducted a blinded user study with a domain expert in immunology to compare the interpretability of neurons from scGPT and scVI with concepts derived from SAEs. For each SAE, we randomly selected 20 concepts and for each model, 20 neurons. For every concept and neuron, we identified the 10 cells with the highest activation and computed attribution scores using the methodology described in Section 3.2, with $N_C = 3$ counterfactual perturbations. The expert was presented with these elements via the interactive visualization interface introduced in Section 3.3, and responded to a set of questions assessing interpretability. Screenshots of the visualizations and evaluation form are provided in Appendix D.

Results show that concepts are more interpretable than neurons, both for scGPT and scVI (Figure 3.A.a). Examples of interpretable concepts are given in Figure 3.B. Several concepts correspond to specific cell types, such as "Myocytes" (concept C3291 from scGPT) and "cytotoxic lymphocyte" (concept C102 from scVI). Other concepts correspond to biological processes, such as "Chemotaxis. Secretion of chemokines" (concept C23 from scVI).

Due to resource constraints, the user study involved only a single participant, which has certain limitations. To assess intra-user consistency, we had the participant re-annotate a subset of concepts. We observed that the "Signal but unclear" annotation is unstable, with some concepts switching between "Not interpretable" and "Signal but unclear." In fewer cases, concepts also changed from a positive annotation to "Not interpretable" and vice versa. Additional annotations from other participants would help strengthen these results.

**Interpretation with pathway enrichment.** We computed pathway enrichment as described in Section 3.4. We selected the pathway with the highest enrichment score, and distinguished weak

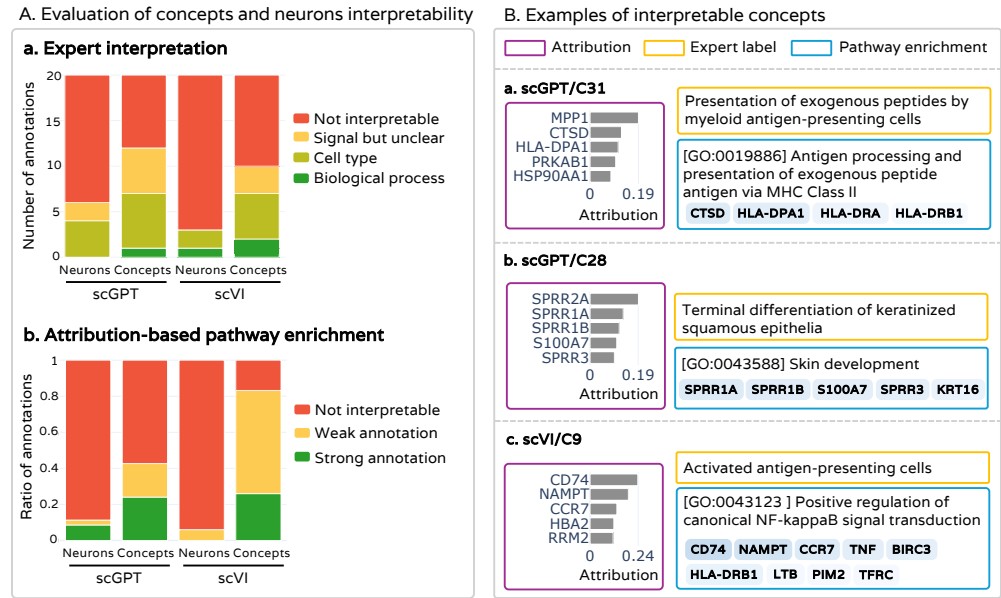

Figure 3: **Concept interpretation results.** (A) Interpretability of concepts compared to neurons. (a) Interpretations of neurons and concepts by a domain expert; (b) Interpretation of neurons and concepts with attribution-based GSEA. Strong annotation corresponds to enriched pathways with p-value $\leq$ 5e-5 (p-value $\leq$ 5e-3 for weak annotations). (B) Examples of interpreted concepts.

annotations ($p - value < 5e - 3$) from strong annotations ($p - value < 5e - 5$) (Figure 3.A.b). Similar to the expert-based interpretation study, the results show that concepts are more aligned with pathways compared to neurons. Examples of pathway enrichment results are given in Figure 3.B.

Despite being more interpretable than neurons, nearly half of the concepts could not be interpreted by either the expert or the pathway enrichment method. Some of these concepts may remain uninterpretable due to the limitations in current biological knowledge. Additionally, the domain expert relied on external resources not yet integrated into our platform [4]. Seamless integration of this prior knowledge could facilitate the interpretation of a greater number of concepts and enable the discovery of more subtle signals.

### 4.4 COMPARISON BETWEEN DATASETS REVEALS STABLE CONCEPTS

Several works showed an instability issue of SAEs, where SAEs trained on different datasets or with different seeds extract different concepts (Fel et al., 2025a; Paulo & Belrose, 2025), which questions their reliability. We trained two sets of Topk SAEs, one on the Tabula Sapiens Immune dataset and the other on the Cross-Tissue Immune Cell Atlas. Since both datasets contain immune cells, we expect that the concepts identified by the first set of SAEs overlap, at least to some extent, with those extracted by the others.

**SAEs generalize to unseen dataset.** We first evaluated whether SAEs trained on a given dataset ("training dataset") could reconstruct cell embeddings from another dataset ("test dataset"), given the $R^2$ score. For all SAEs, the $R^2$ is slightly lower for the test dataset compared to the training dataset (Figure 4.A), suggesting that some concepts specific to the test dataset may be missing. Especially, the gap is smaller for SAEs trained on the Tabula Sapiens Immune dataset, which contains approximately twice as many cells as the Cross-tissue Immune Cell Atlas. A more pronounced drop in $R^2$ is observed for scGPT, which we hypothesize is due to differences in the input gene sets between the two datasets, with only 250 genes in common.

---

[4]The Protein Atlas (Uhlén et al., 2015) and articles from PubMed https://pubmed.ncbi.nlm.nih.gov/

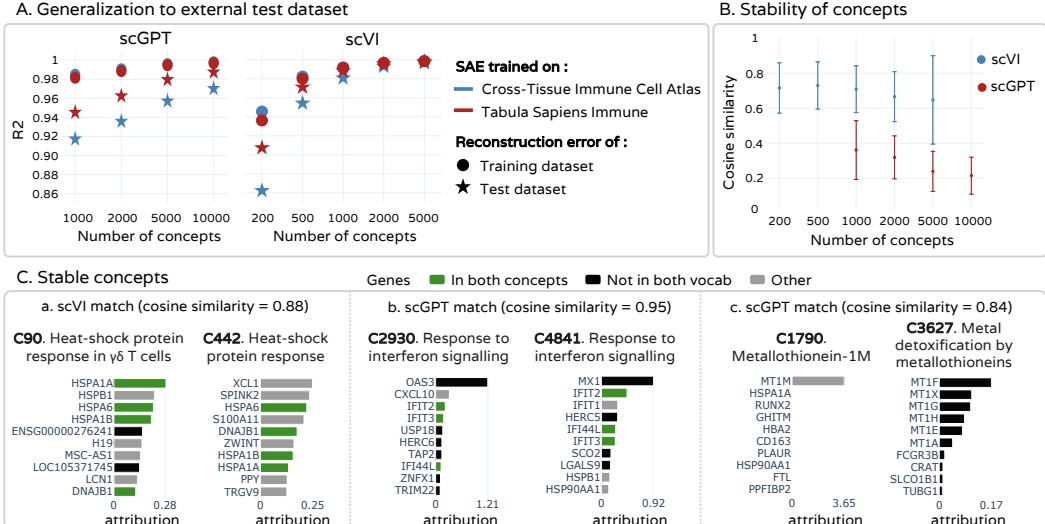

Figure 4: **Stability of SAEs trained on different datasets**. (A) Reconstruction error of cell embeddings from an external dataset compared to training samples. (B) Cosine similarity of matched concept vectors from SAE trained on the Tabula Sapiens Immune and SAE trained on the Cross-tissue Immune Cell Atlas, after finding the best alignment via the Hungarian algorithm as proposed in Fel et al. (2025a). (C) Examples of matching concepts with their most important genes. For each pair, the concept on the left is from the SAE trained on Tabula Sapiens Immune, and the concept on the right is from the SAE trained on the Cross-Tissue Immune Cell Atlas.

**Some concepts are stable between SAEs.** We used the method introduced in Fel et al. (2025b) to match concept vectors from SAEs trained on the Tabula Sapiens Immune dataset and SAEs trained on the Cross-Tissue Immune Cell Atlas. It matches pairs of concepts by minimizing the cosine distance between concept vectors ($D$) with the Hungarian algorithm. The cosine similarity of the obtained matching indicates how well the two SAEs align. A score of 1 means that the two dictionaries are identical up to a permutation. Concepts extracted from scVI embeddings are more stable compared to scGPT, which is on par with the generalization results (Figure 4.B). We further explored pairs of concepts with a high cosine similarity. These concepts often have a few genes in common among the 10 most important genes. More interestingly, even if important genes do not perfectly intersect, the concepts share a common interpretation (Figure 4.C). In particular, one pair of concepts does not have any top-10 genes in common, but the genes in the two concepts are from the same family (Figure 4.C.d).

## 4.5 Towards useful concept representations

In this experiment, we evaluated the usefulness of concepts for interpretable downstream tasks using two classification problems: cell cycle phase (3 classes: G1, S, G2M) and cell type (7 selected classes). We trained logistic regression models on either concept activations or neuron activations and validated concepts having a high coefficient using known marker genes for the task. The experiment was conducted with scGPT and the Tabula Sapiens Immune dataset. We obtained cell cycle phase labels using Scanpy [5] (Wolf et al., 2018). Details are provided in Appendix G.

For both tasks, models trained on concept activations achieve similar accuracy to those trained on neuron activations (respectively 0.86 and 0.87 for cell type and 0.49 and 0.48 for cell cycle phase). Concept activations, hence, preserve the signal. We further explored the coefficients of the logistic regression models for cell cycle phase classification and found that concepts with high coefficients are relevant for the task. The genes characterizing the concepts are mainly gene markers for the G2M phase (Figure 5.A.2). In comparison, neurons with high coefficients in logistic regression do not appear relevant for the task (Figure 5.B). Interestingly, concept C2022, labeled as "active mi-

---
[5]Scanpy tool "score_genes_cell_cycle"

totic program", positively predicts G2M phase and negatively predicts S phase. The other concepts display mixed signals associated with both phases, suggesting that cell cycle information may not be linearly encoded in the latent space. This could explain the limited predictive performance of logistic regression models on this task.

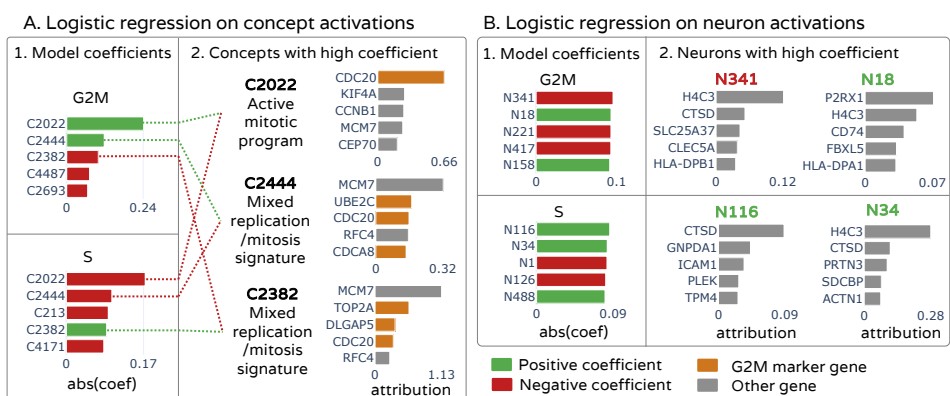

Figure 5: **Interpretation of cell cycle phase classification.** (A) Key concepts contributing to predictions based on concepts. (B) Key neurons contributing to predictions based on neurons.

## 5 RELATED WORK

**Explainability of scRNA-seq models**   Interpretability in scRNA-seq models often relies on pathway enrichment methods (Maleki et al., 2020), which interpret the model's mechanism through the lens of existing biological knowledge encoded in curated ontologies such as Reactome and Gene Ontology (Fabregat et al., 2016; Ashburner et al., 2000). Some approaches incorporate prior biological knowledge directly into model architecture, designing models in which certain components correspond to known biological processes (Bourgeais et al., 2022; Rybakov et al., 2020; Lotfollahi et al., 2023; Gut et al., 2021; Zarlenga et al., 2024; Ruiz-Arenas et al., 2024; de la Fuente et al., 2025). Although this strategy yields models that are interpretable by design, it also constrains them to existing knowledge, thereby limiting their potential for discovery. In contrast, post-hoc explainability aims to interpret models after training. Attribution methods are frequently used to identify the genes that contribute most to specific predictions (Yap et al., 2021; Usman et al., 2025; Chereda et al., 2021). For comprehensive overview, Zhou et al. (2023) and Conard et al. (2023) review explainable and interpretable machine learning methods for biology. Our work falls within the post-hoc approaches and provides new tools to interpret any black-box neural network already trained on single-cell RNA-seq data.

**Sparse dictionary learning for interpretability of deep learning models in biology**   Sparse dictionary learning has recently shown great potential for decomposing the latent space of deep learning models into sparse and interpretable features. Following its success in language model (Sharkey et al., 2022; Huben et al., 2023) and vision models (Fel et al., 2023), this methodology has been extended to deep learning models for biology. Sparse Auto-Encoders (SAEs) have successfully uncovered meaningful concepts encoded by protein language models, such as generic and family-specific features (Adams et al., 2025), or binding sites and structural motifs (Simon & Zou, 2024). SAEs have also been applied to histopathology models, where they discovered interpretable concepts related to cellular and tissue characteristics, and geometric structures (Le et al., 2024). Alongside our work, Schuster (2024) trained a Sparse Auto-Encoder on the cell embeddings from a scRNA-seq generative model and used pathway enrichment to map scRNA-seq concepts to known pathways. We introduce different interpretation methods that go beyond correlational approaches and conduct a user study. Additionally, we assess the stability and usefulness of the resulting concepts, which are necessary for practical utility.

# 6   CONCLUSION

This work introduces a comprehensive framework for interpreting biological concepts in scRNA-seq foundation models using sparse autoencoders. We addressed key challenges in scRNA-seq concept interpretation by proposing a principled approach to identify genes that influence concept activation, and an interactive visualization tool that integrates prior knowledge. By collaborating with a domain-expert, we were able to interpret biologically meaningful concepts and demonstrate that SAEs can extract concepts that are more interpretable than individual neurons. We further showed that concept activations preserve the biological signal of the original representations and identified concepts that are stable across independent datasets. Several important directions emerge from this work. The integration of richer biological prior knowledge, either at the extraction or interpretation stage, such as biological knowledge graphs or regulatory networks, could improve alignment with current biological knowledge and enable the interpretation of more subtle signals. Additionally, the concept space learned by SAEs offers natural intervention points for controlling model behavior through steering, possibly supporting applications such as perturbation response prediction and exploration of counterfactual biological scenarios.

## REPRODUCIBILITY STATEMENT

We will release our visualization platform after the review period, which will allow readers to explore all the computed results from this work. Section 4.1 and Appendix A describe models and datasets used in this work, all of which are openly available. Section 4.2 and Appendix B describe the training setup for Topk SAEs. Section 3 describes in detail the method that we used to interpret concepts in Section 4.

## ETHICS STATEMENT

In preparing this manuscript, we occasionally used suggestions from LLMs (GPT-5) to guide improvements in clarity, grammar, and overall readability. All scientific content, including experimental design, data analysis, results, and interpretations, is independently developed by the authors.

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

## A    DATASETS AND MODELS

The two datasets used in this work, the Cross-tissue Immune Cell Atlas (Domínguez Conde et al., 2022) and Tabula Sapiens Immune (Consortium* et al., 2022), are described in Table 1.

Table 1: Description of scRNA-seq datasets.

|  | **Cross-tissue Immune Cell Atlas** | **Tabula Sapiens Immune** |
|---|---|---|
| # cells | 329 762 | 592 317 |
| # genes | 36 601 | 61 759 |
| # genes after pre-processing | 36 079 | 61 757 |
| # patients | 12 | 28 |
| # tissues | 17 | 74 |
| # cell types | 45 | 45 |

The two models used in this work, scVI (Lopez et al., 2018) and scGPT (Cui et al., 2024) are described in Table 2. UMAP visualizations of cell embeddings from these models are displayed in Figures 6 and 7.

Due to discrepancies between the gene names in the Cross-tissue Immune Cell Atlas and in the scVI vocabulary, only 3896/8000 genes had a match for this dataset. This most certainly alters the results for this combination of model and dataset.

Table 2: Description of scRNA-seq foundation models.

|  | **scVI** | **scGPT** |
|---|---|---|
| Model architecture | VAE | Transformer |
| Encoder parameters (total) | 2M (8M) | 50M (100M) |
| Genes in vocabulary | 8 000 | 60 698 |
| Gene expression preprocessing | sum norm + log1p | binning |
| Sequence length | 8000 HVG | 1200 HVG |
| Cell embedding strategy | Latent embedding | CLS token |
| Cell embedding size | 50 | 512 |

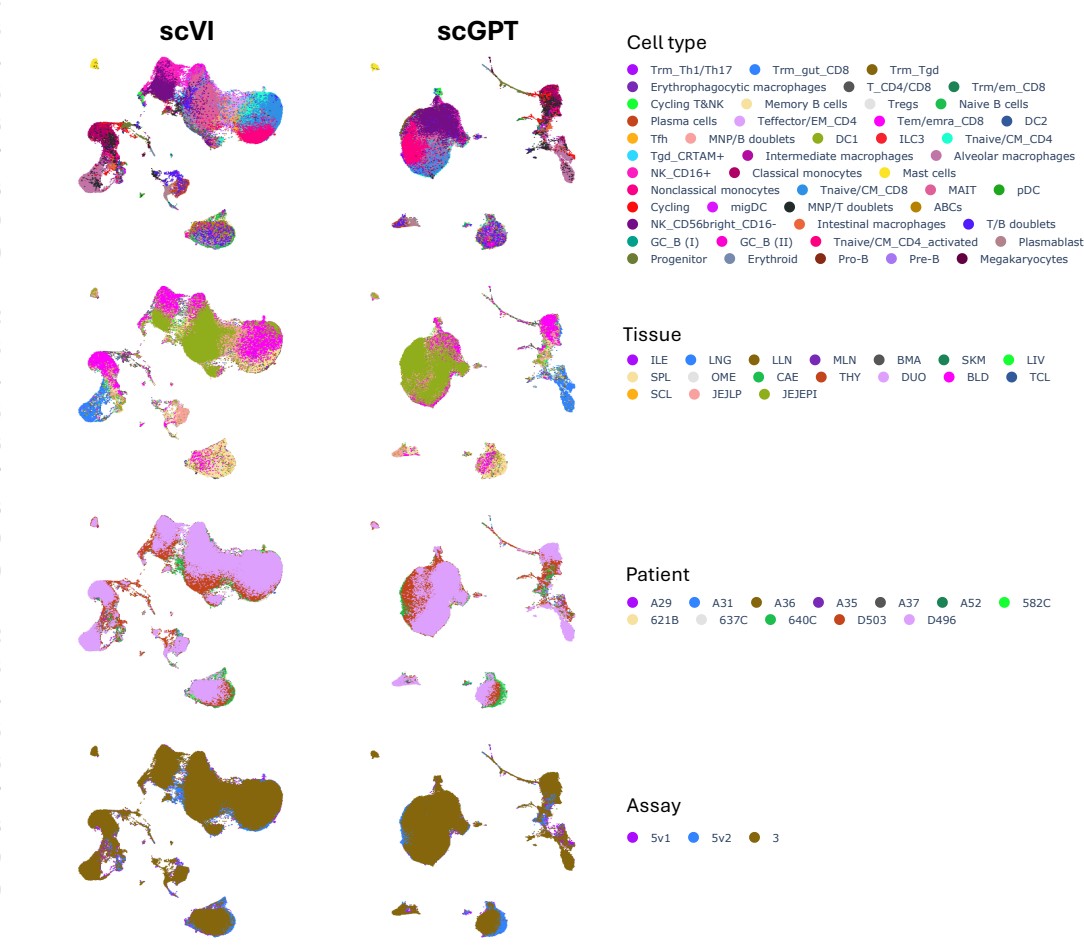

Figure 6: UMAP of cell embeddings from the Cross-Tissue Immune Cell Atlas (Domínguez Conde et al., 2022), colored by metadata.

## B   TOPK SAEs

**Hyper-parameter tuning**   Hyper-parameters and metrics for the different SAEs are given in Table 3. Learning rate seems to play an important role in the number of dead concepts at the end of training. We mainly tuned this parameter. Batch size is fixed to 1024 and $aux\_k$ to 512.

**Comparison with semi-NMF**   Other decomposition methods can be used for concept extraction, such as Non-Negative Matrix Factorization (NMF). Previous works have shown that Sparse Auto-Encoders scale better to large datasets and have better sparsity-reconstruction trade-off (Fel et al., 2025b). We conducted experiments on single-cell RNA-seq data and found similar results. We compared TopK SAEs to semi-NMF[6], a relaxed version of NMF suited to activations that are not non-negative, in both undercomplete ($c < d$) and overcomplete ($c > d$) settings. Results are given in Table 4 for scGPT and Table 5 for scVI. For the same number of concepts $c$ and comparable sparsity levels, the decomposition approach has weaker reconstruction performance: $R^2$ of 0.980 (TopK SAE) vs. 0.849 (semi-NMF) for scVI, and 0.995 (topK SAE) vs. 0.933 (semi-NMF) for scGPT. We also evaluated the preservation of biological signal, using the cell cycle phase and cell type classification tasks described in Section 4.5 and Appendix G. While TopK SAE concepts closely match the performance of neurons, the accuracy decreases with semi-NMF concepts: for cell type

---

[6]Using the code from the Overcomplete library : https://github.com/KempnerInstitute/overcomplete

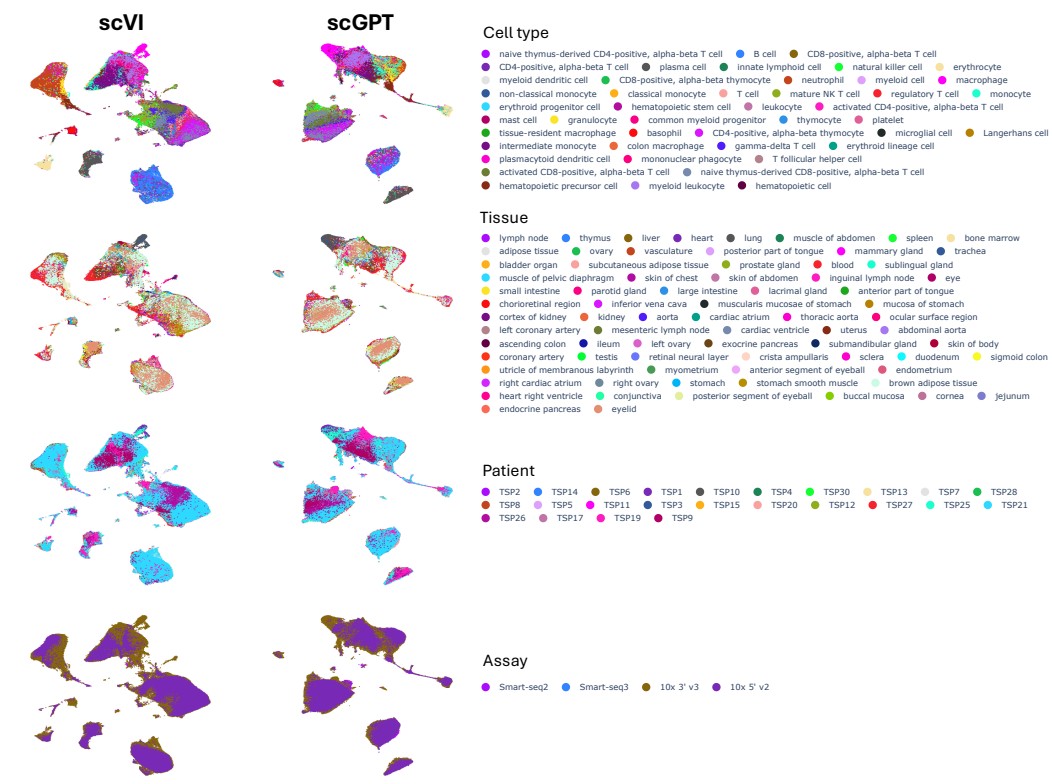

Figure 7: UMAP of cell embeddings from Tabula Sapiens Immune (Consortium* et al., 2022), colored by metadata.

Table 3: Training hyper-parameters and metrics for Topk SAEs.

| Dataset | Model | $c$ | $k$ | lr | Epochs | R2 | Active concepts |
|---------|-------|-----|-----|-----|--------|-----|-----------------|
| Cross-tissue Immune Cell Atlas | scGPT | 1000 | 16 | 1e-4 | 1000 | 0.985 | 811 |
| Cross-tissue Immune Cell Atlas | scGPT | 2000 | 32 | 5e-5 | 2500 | 0.990 | 1761 |
| Cross-tissue Immune Cell Atlas | scGPT | 5000 | 80 | 5e-5 | 2500 | 0.996 | 3728 |
| Cross-tissue Immune Cell Atlas | scGPT | 10000 | 160 | 5e-5 | 1850 | 0.998 | 6923 |
| Cross-tissue Immune Cell Atlas | scVI | 200 | 3 | 1e-4 | 2000 | 0.946 | 198 |
| Cross-tissue Immune Cell Atlas | scVI | 500 | 8 | 1e-4 | 2000 | 0.982 | 487 |
| Cross-tissue Immune Cell Atlas | scVI | 1000 | 16 | 1e-4 | 2000 | 0.992 | 974 |
| Cross-tissue Immune Cell Atlas | scVI | 2000 | 32 | 5e-4 | 2000 | 0.997 | 1607 |
| Cross-tissue Immune Cell Atlas | scVI | 5000 | 80 | 5e-4 | 4000 | 0.999 | 3399 |
| Cross-tissue Immune Cell Atlas | scVI | 10000 | 160 | 5e-4 | 3000 | 0.998 | 8803 |
| Tabula Sapiens Immune | scGPT | 1000 | 16 | 1e-4 | 600 | 0.981 | 964 |
| Tabula Sapiens Immune | scGPT | 2000 | 32 | 7e-5 | 1200 | 0.988 | 1887 |
| Tabula Sapiens Immune | scGPT | 5000 | 80 | 5e-5 | 1500 | 0.995 | 4381 |
| Tabula Sapiens Immune | scGPT | 10000 | 160 | 5e-5 | 1092 | 0.996 | 8799 |
| Tabula Sapiens Immune | scVI | 200 | 3 | 1e-4 | 1000 | 0.936 | 199 |
| Tabula Sapiens Immune | scVI | 500 | 8 | 1e-4 | 1500 | 0.988 | 489 |
| Tabula Sapiens Immune | scVI | 1000 | 16 | 1e-4 | 1000 | 0.991 | 972 |
| Tabula Sapiens Immune | scVI | 2000 | 32 | 5e-4 | 2000 | 0.997 | 1634 |
| Tabula Sapiens Immune | scVI | 5000 | 80 | 5e-4 | 3000 | 0.998 | 4108 |
| Tabula Sapiens Immune | scVI | 10000 | 160 | 5e-4 | 1300 | 0.999 | 8790 |

classification, accuracy of 0.85 (topK SAE) vs. 0.79 (semi-NMF) for scVI, and 0.86 (topK SAE) vs. 0.73 (semi-NMF) for scGPT, indicating a loss of biological signal. We also evaluated the low-rank

setting with $c < d$ and found a marked decrease in reconstruction performance and downstream tasks accuracy.

| | Neurons | TopK SAE | Semi-NMF ($c > d$) | Semi-NMF ($c < d$) |
|---|---|---|---|---|
| Number of concepts $c$ | 512 | 5000 | 5000 | 200 |
| Active concepts | – | 4381 | 5000 | 196 |
| Sparsity | – | 0.984 (0.0) | 0.971 (0.011) | 0.979 (0.009) |
| Reconstruction (R2) | – | 0.995 (0.003) | 0.933 (0.032) | 0.858 (0.062) |
| Cell cycle (accuracy) | 0.482 | 0.487 | 0.477 | 0.436 |
| Cell type (accuracy) | 0.869 | 0.860 | 0.734 | 0.300 |

Table 4: Comparison of concept extraction methods for scGPT ($d = 512$), results on the Tabula Sapiens Immune dataset. Cell cycle and cell type tasks correspond to the tasks introduced in Section 4.5. Semi-NMF with $c = 5000$ was fitted on 50% of the training set instead of 90% for the other methods, due to memory issues.

| | Neurons | TopK SAE | Semi-NMF ($c > d$)) | Semi-NMF ($c < d$) |
|---|---|---|---|---|
| Number of concepts $c$ | 50 | 500 | 500 | 20 |
| Active concepts | – | 489 | 496 | 20 |
| Sparsity | – | 0.984 (0.0) | 0.989 (0.003) | 0.830 (0.057) |
| Reconstruction (R2) | – | 0.980 (0.014) | 0.849 (0.110) | 0.713 (0.171) |
| Cell cycle (accuracy) | 0.493 | 0.485 | 0.468 | 0.462 |
| Cell type (accuracy) | 0.853 | 0.853 | 0.779 | 0.692 |

Table 5: Comparison of concept extraction methods for scVI ($d = 50$), results on the Tabula Sapiens Immune dataset. Cell cycle and cell type tasks correspond to the tasks introduced in Section 4.5.

## C DIFFERENTIAL GENE EXPRESSION ANALYSIS (DGEA)

**DGEA-based gene set** We compare genes identified with the attribution method that we propose to genes identified with Differential Gene Expression Analysis as proposed by Schuster (2024). Differential gene expression analysis is performed between cells that maximally activate the concept (with a maximum of 1000 cells) and counterfactual cells (with a maximum of 1000 cells).

**Deletion curve** We computed deletion curves to compare the impact on concept activation of the two sets of genes (Figure 8). Genes are sorted by attribution value for the attribution-based deletion, and sorted by absolute FoldChange for DGEA-based deletion (only for genes with $p - value \leq 5e - 3$). We then perturb cells with counterfactual perturbations, from the most important to the least important, and compute the perturbed concept activation. Attribution-based deletion curve is below DGEA-based deletion curve, which demonstrates that genes obtained via attribution have a bigger impact on concept activation. We note that gene perturbations have a greater impact on the embeddings of scGPT cells compared to scVI. We will investigate this behavior in future work.

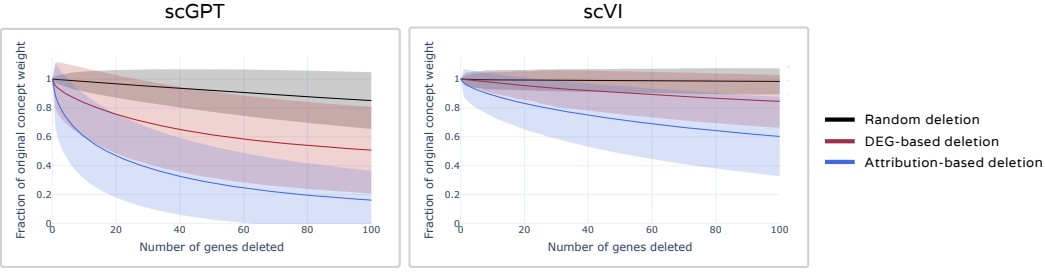

Figure 8: Deletion curve for all concepts (mean and std at each gene deletion step)

**Comparison of attribution-based GSEA and DGEA-based GSEA**  We computed DGEA-based GSEA and attribution-based GSEA of the first 100 concepts of the TopK SAEs used in Section 4.3. For a given concept, we obtain two sets of biological processes : $P_{att} = \{T_1^{att}, ..., T_{k_{att}}^{att}\}$ the biological processes from attribution-based GSEA and $P_{dgea} = \{T_1^{dgea}, ..., T_{k_{dgea}}^{dgea}\}$ the biological processes from DGEA-based GSEA. Each biological process $T$ is defined as a set of genes $T = \{(g_1, FC_1, att_1), ..., (g_m, fc_m, att_m)\}$ with $g$ the gene name, $fc$ the log2 fold change from DGEA and $att$ the attribution score. We compared the results only if there is at least one biological process in both $P_{att}$ and $P_{dgea}$. The metrics are :

- The maximal absolute log2 fold change value of the genes in the biological processes: $\max_{T \in P_{att}} \ \max_{(g, \text{fc}, \text{att}) \in T} |\text{fc}|$

- The maximal attribution score of the genes in the biological processes: $\max_{T \in P_{att}} \ \max_{(g, \text{fc}, \text{att}) \in T} \text{att}$

- The intersection of biological processes : $\frac{|P_{\text{att}} \cap P_{\text{dgea}}|}{|P_{\text{att}} \cup P_{\text{dgea}}|}$

- The intersection of genes in the biological processes: $\frac{|G_{\text{att}} \cap G_{\text{dgea}}|}{|G_{\text{att}} \cup G_{\text{dgea}}|}$ with $G_\star = \bigcup_{T \in P_\star} \{g \mid (g, \text{fc}, \text{att}) \in T\}$

The results are given in Table C. First, we observe minimal overlap between the biological processes detected by two methods (mean IoU of 0.068 for scGPT and 0.025 for scVI) and between the genes within these processes (mean IoU of 0.055 for scGPT, 0.027 for scVI), demonstrating the need to choose one of the methods. As expected, biological processes identified through classic DGEA-based GSEA contain genes with higher absolute log2 fold change (mean 4.0 vs. 2.6 for scGPT, 4.7 vs. 2.4 for scVI), whereas the biological processes identified with the attribution-based method contain genes with higher attribution scores (mean 0.32 vs. 0.17 for scGPT, 0.21 vs. 0.14 for scVI). Deletion curves (Figure 8) further indicate that genes identified via attribution exert a greater impact on concept activation than those identified via DGEA. Together, these results justify the use of attribution-based GSEA for concept interpretation, as the resulting biological processes more accurately reflect the signal associated with the concept.

| | scVI | | scGPT | |
|---|---|---|---|---|
| | attribution-based | DGEA-based | attribution-based | DGEA-based |
| Max absolute log2 fold change | 0.211 (0.215) | 4.688 (2.211) | 0.323 (0.405) | 4.040 (1.837) |
| Max attribution | 2.362 (1.812) | 0.135 (0.169) | 2.570 (2.261) | 0.172 (0.407) |
| Number of concepts enriched | 72 | 89 | 32 | 62 |
| IoU genes | 0.027 (0.043) | | 0.055 (0.099) | |
| IoU biological processes | 0.025 (0.060) | | 0.068 (0.138) | |

Table 6: Comparison of DGEA-based GSEA and attribution-based GSEA for 100 concepts per model. We only consider enriched biological processes with p-value $\leq 0.005$.

## D  INTERFACE FOR THE EXPERT INTERPRETATION STUDY

Screenshots of the interface that we developed and deployed for the expert interpretation study are provided in Figure 9.

## E  LIMITS OF PATHWAY ENRICHMENT

There are several limitations to GSEA interpretation based on the biological processes of the Gene Ontology. First, many genes seem to play a role in the activation of concepts but are not mapped to any biological process yet (270/1200 genes for scGPT and 5113/8000 genes for scVI). The tree structure of the ontology can also generate confusion because sister terms (GO terms from the same parents) can be either close or very dissimilar in their semantic meaning, and this caveat is of partic- ular importance when neglecting the graph structure of the Gene Ontology and ignoring the type of edges linking biological processes. The inherent incompleteness of gene annotation and the ontol- ogy itself leads to the so-called "streetlight effect" skewing the interpretation towards what is known

Interactive visualization of a concept          Form with questions

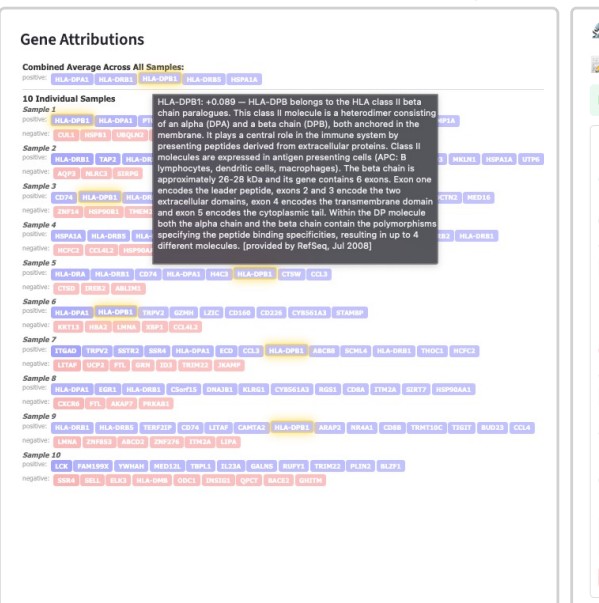
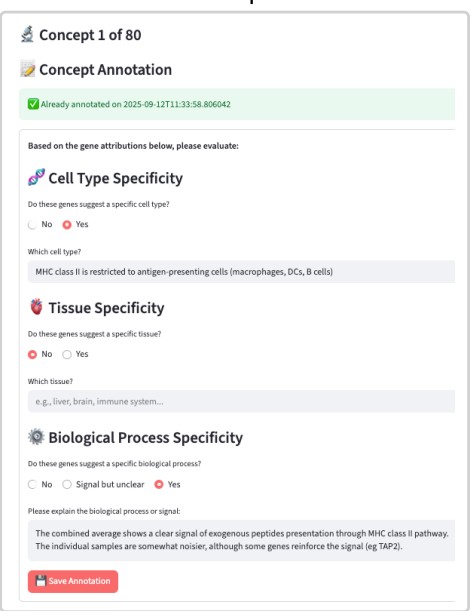

Figure 9: Screenshots of the interface for the expert interpretation study. (Left) Interactive visualization of a concept. (Right) Form with questions asked to the expert.

rather than where truth really is. This leads to a heterogeneous ontology, with an uneven depth of annotation, which can bias enrichment assays.

## F  INTERPRETED CONCEPTS

We provide complete information for a few concepts interpreted in Section 4 in Table 7. In particular, we provide an additional description of the concepts.

## G  DOWNSTREAM TASKS

**Cell cycles**  We labeled the Tabula Sapiens Dataset with cell cycle phase using the Scanpy tool "score_genes_cell_cycle" (Wolf et al., 2018). We then built a balanced dataset with the three classes: "G2M", "S", and "G1". The training dataset comprises 10000 samples per class, and the test dataset comprises 1000 samples per class. We used the logistic regression from sklearn with default parameters. Accuracy is similar for classification from concept activations and neuron activations (Figure 10.B). Concepts and neurons with the highest coefficient are given in Figure 5. Markers genes are the one provided with the Scanpy code [7]. Note that only 8 out of 97 of these genes are in the input of the model.

**Cell types**  We selected 7 classes with enough samples to balance the dataset : "B cell", "CD8-positive, alpha-beta T cell", "CD4-positive, alpha-beta T cell", "natural killer cell", "neutrophil", "classical monocyte", "monocyte". The training dataset comprises 6000 samples per class, the test dataset comprises 2000 samples per class. We use the logistic regression from sklearn with default parameters and $max\_iter = 50$. Accuracy is similar for classification from concept activations and neuron activations (Figure 10.A).

---

[7]https://github.com/scverse/scanpy_usage/blob/master/180209_cell_cycle/data/regev_lab_cell_cycle_genes.txt

Table 7: Interpretable concepts.

| Concept ID | Expert label | Expert description | Pathway enrichment label | Top-10 important genes |
|---|---|---|---|---|
| scGPT/C28 | Terminal differentiation of keratinized squamous epithelia | Genes with the highest attribution values are either key contributors to the structural formation of the cornified layers in tissues such as the outer part of the skin, and/or part of the epidermal differentiation complex, a group of genes central to terminal differentiation of the skin epithelial cells | Keratnocyte differentiation (GO:0030216) | SPRR2A, SPRR1A, SPRR1B, S100A7, SPRR3, HSP90AA1, KRT13, SBSN, KRT16, MKLN1 |
| scGPT/C31 | Presentation of exogenous peptides by myeloid antigen-presenting cells | The gene set contains several genes linked to the presentation of exogenous molecules at the surface of our cells. A set of genes is evocative of macrophages, which are professional antigen-presenting cells, while CTSD relates to their capacity to digest exogenous molecules before presenting them | Antigen processing and presentation of exogenous peptide antigen (GO:0002478) | MPP1, CTSD, HLA-DPA1, PRKAB1, HSP90AA1, PPP1CC, HLA-DRA, SERPINA1, DCTN2, ZNF585A |
| scVI/C9 | Activated antigen-presenting cells | Although it relates to the concept presented above, they are not identical. While the core machinery of exogenous antigen presentation is also present, a set of genes suggests a focus on activated antigen-presenting cells rather than the presentation process itself | Postitive regulation of canonical NF-KappaB signal transduction (GO:43123) | CD74, NAMPT, CCR7, HBA2, RRM2, ALB, TNF, BIRC3, HLA-DPB1, CD86 |
| scVI/C13 | Plasmacytoid dendritic cells | The antigen-presentation machinery is again included here, together with specific transcription factors and effector molecules. While some genes are discordant with this interpretation, the concept likely relates to plasmacytoid dendritic cells | Positive regulation of immune response (GO:0050778) | GZMB, FGFBP2,CD74, CDK6, ENSG00000288891, HLA-DRA, HSPH1, KCNQ5, MTCO1P12, AURKB |

# H    ATTRIBUTION

Cumulative distribution of attribution scores are given in Figure 11.

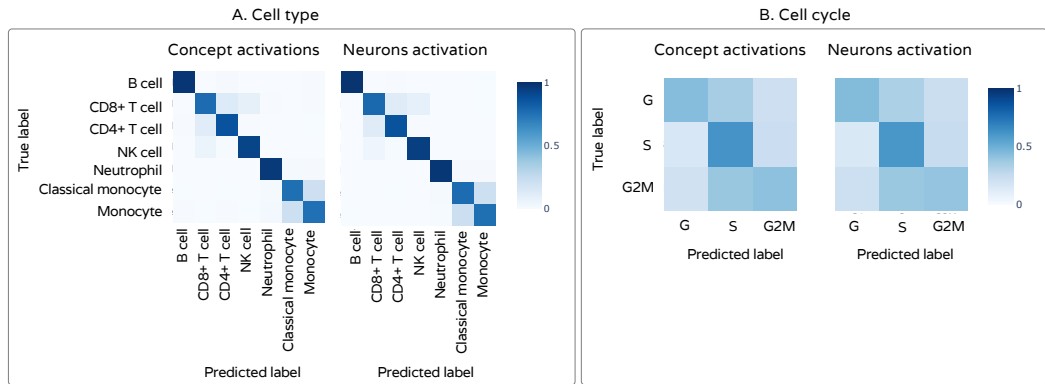

Figure 10: Confusion matrix of predictions on test sets for the two tasks. (A) Cell type classification. The test set contains 2000 samples per class, the accuracy of predictions from concept activations is 0.86 and 0.87 from neuron activations. (B) Cell cycle classification. The test set contains 1000 samples per class, the accuracy of predictions from concept activations is 0.49 and 0.48 from neuron activations.

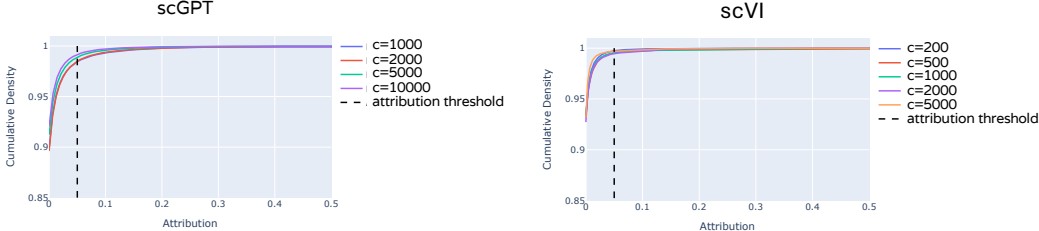

Figure 11: Cumulative distribution of attribution scores.

