# OpenReview forum: "Discovering interpretable biological concepts in single-cell RNA-seq foundation models"
_ICLR.cc/2026/Conference — Submitted to ICLR 2026_

### Official Review · Reviewer_Tc45 · 2025-10-26

**Soundness:** 3
**Presentation:** 3
**Contribution:** 2
**Rating:** 6
**Confidence:** 3

**Summary:**

This paper introduce an interpretability framework for single-cell RNA-seq models based on concept extraction through a decomposition approach, which relies on sparse auto-encoders. To evaluate these, they propose an attribution-based method with conterfactual perturbations, an attribution-based gene set enrichment analysis (GSEA) and a web-based visualization tool with an expert interpretation. Experiments on Tabula Sapiens Immune and Cross-Tissue Immune Cell Atlas datasets show that these concepts are more interpretable and stable than individual neurons while retaining downstream predictive power.

**Strengths:**

1. Paper is well written and structured.

2. Addresses interpretability of large scRNA-seq foundation models, which is still an area of high interest.

3. Combines sparse dictionary learning with counterfactual-based attributions, going beyond correlation-based analyses.

4. Uses large immune cell datasets; includes cross-dataset stability analysis.

5. Incorporates an area expert, which adds credibility and biological realism.

**Weaknesses:**

1. The novelty of the proposed top-k sparse auto-encoder to extract meaningful concepts is limited.

2. Expert evaluation involves only one annotator; inter-expert variability or reproducibility is not assessed.

3. The “interpretability gain” is mostly qualitative, no standardized metric is provided.

4. Both datasets are immune-cell focused; generalization to other tissues or modalities remains untested.

**Questions:**

1. Is the code and visualization platform available at submission? I can't find it in the paper.

2. Could you highlight the main difference between other models that provide an interpretable latent space such as VEGA (https://doi.org/10.1093/bioinformatics/btad387), GSAE (https://doi.org/10.1186/s12918-018-0642-2) or SENA-discrepancy-VAE (https://openreview.net/pdf/f8c0bd3ea256d2af7cfe66c6a46b511cab5ba995.pdf)? Couldn't we forward the single-cell samples through any of these and extract similar features?

2. How does the attribution-based GSEA differ quantitatively from traditional DGEA-based enrichment? Can you provide examples of pathways uniquely discovered?

3. How stable are concepts when changing SAE random seeds or hyperparameters? Could you include std in metrics where multiple seeds have been tested?

---

> ### Author Response · Authors · 2025-11-21
>
> Dear reviewer, we greatly appreciate your positive comments on the clarity and structure of the paper. Thank you for your feedback on weaknesses and for your questions, which will help us further enhance the manuscript. Please find our responses and additional experiments below.
>
> **W1**: The concept extraction method is indeed not the novelty of the paper. While concept extraction has been widely explored in the literature, there are still few approaches that interpret concepts specifically in biological settings and for complex data such as single-cell RNA-seq. Our contribution lies in introducing methods to interpret single-cell RNA-seq concepts and assess the biological knowledge encoded by well-known models.
>
> **W2**: We acknowledge the limitations of the study, which involves a single annotator. Please find an additional experiment on inter-expert variability in the common answer (number 2).
>
> **W3**: We quantitatively assessed the interpretability gain by comparing the interpretability of concepts compared to neurons. We agree that the metrics focus on individual concepts' interpretability. Were you interested in other aspects? Can you provide examples of standardized metrics you would expect?
>
> **W4**: While the datasets contain only immune cells, the cells were collected in several tissues. We agree that the datasets limit the scope of the study; however, the interpretation methods that we introduce are not specific to immune cells and could be applied to any other dataset. The main hyperparameter to tune would be the number of concepts $c$, which we expect to be larger for more diverse datasets.
>
> **Q1**: The visualization platform is not currently available and will be released along with the paper due to privacy concerns during the review.
>
> **Q2**: The need to clarify the positioning was also mentioned by other reviewers, please find our answer in the common answer (number 1).
>
> **Q3**: Other reviewers also requested additional experiments with traditional DGEA-based enrichment. Please find the detailed answer in the common answer (number 3).
>
> **Q4**: All experiments were conducted at a fixed seed, we did not test several seeds due to compute budget. We agree that this is a limitation, please find the results of additional experiments on the seed and hyperparameter k, using the stability metric used in the paper :
>
> We trained SAEs with 3 different seeds and compared to the SAE used in the paper:
> - scVI (with c=500 and k=8): for all seeds we obtain a stability score of 0.79 +/- 0.12
> - scGPT (with c=5000 and k=80): for all seeds we obtain a stability score of 0.38 +/- 0.17
>
> Cosine similarity in high dimensions is hard to interpret; however, we can say that concepts are more stable when SAEs are trained on different seeds compared to SAEs trained on different datasets, which is expected. While running these additional experiments we identified an error in the figure 4.B in the paper, with the y axis being wrong. We corrected the figure.
>
> We also trained SAEs with 2 other $k$ and computed the stability metric, the hyperparameter $k$ has limited impact on the stability metric (when compared to the seed experiment):
> - scVI (with c=500): 0.79 +/- 0.13 (k=6 vs k=8) and 0.80 +/- 0.11 (k=10 vs k=8)
> - scGPT (with c=5000): 0.36 +/- 0.17 (k=60 vs k=80) and 0.36 +/- 0.18 (k=100 vs k=80)

---

> > ### Comment · Reviewer_Tc45 · 2025-11-25
> >
> > Dear authors,
> >
> > Thank you for carefully addressing most of the concerns I raised in my previous review. While I appreciate the substantial improvements made to the manuscript, I believe that the limited robustness of the expert validation and the lack of sufficient methodological novelty prevent me from increasing the score.

---

### Official Review · Reviewer_YuVv · 2025-10-28

**Soundness:** 3
**Presentation:** 3
**Contribution:** 2
**Rating:** 4
**Confidence:** 4

**Summary:**

This paper introduces a framework for improving the interpretability of single-cell RNA-seq foundation models. The manuscript is clearly written, the methodology is sound, and the experiments are thorough.

The core of the contribution is the gene attribution method based on counterfactual perturbations. By identifying genes that distinguish cells activating a concept from similar cells that do not, this approach moves beyond simple correlational analysis (like DGEA) to better pinpoint influential genes.

A couple of weaknesses (context within the broader field of interpretable ML, justification of methodological choices, depth of the counterfactual analysis, limitations of the expert study, hypothesis generation claim) should be addressed.

**Strengths:**

The core contribution (gene attribution method based on counterfactual perturbations) is a well-grounded methodological advancement.

The empirical validation is sound: The authors provide evidence that concepts extracted via TopK SAEs are more interpretable than individual neurons from the original models scGPT and scVI. Also, the study evaluates the stability of learned concepts across different datasets, a known challenge for SAEs.

**Weaknesses:**

Context within the broader field of interpretable ML: The introduction frames the problem primarily as one of post-hoc explanation for "black box" models. However, it would benefit from acknowledging and contrasting its approach with the branch of causal representation learning, e.g., discrepancy-VAE (Zhang et al.), SENA (de la Fuente et al.), and GEARS (Roohani et al.). Discussing why a post-hoc concept extraction approach might be preferable or complementary (e.g., applicable to any pre-trained foundation model without architectural changes) would provide a more complete picture of the landscape.

Justification of methodological choices: The manuscript states that TopK SAEs simplify tuning over vanilla SAEs with L1 regularization. This is a practical reason, but the authors should provide a more thorough justification for choosing this specific dictionary learning architecture over other methods for learning disentangled representations. For instance, why not use a VAE with a structured prior (e.g., beta-VAE) to learn the concepts directly from the embeddings?

Depth of the counterfactual analysis: The counterfactual for a sample is defined as the closest sample in the embedding space that does not activate the concept. This is a reasonable heuristic, but it has limitations. The "closest" cell might differ in biologically important ways beyond the concept in question, potentially confounding the attribution scores. The authors should discuss this limitation.

Limitations of the expert study: The study was conducted with a single domain expert. While valuable, this is an N=1 study, and the generalizability could be questioned. The authors should explicitly state this as a limitation and perhaps soften their conclusions slightly. Was there a process to measure intra-expert consistency (e.g., showing the same concept twice)? This should be clarified.

Hypothesis generation claim: The examples show that the framework can recover well-established biological concepts. To fully substantiate the claim of enabling hypothesis generation and discovery, it would be powerful to showcase an example of a concept that is either novel, refines existing knowledge, or reveals a non-obvious connection between genes that could lead to a testable biological hypothesis.

Typos:
- Page 2, line 104: \sigma(\cdot)
- Page 2, line 107: L1 regularization

**Questions:**

The questions directly relate to the identified weaknesses:
- How does the post-hoc concept extraction framework complements or offers advantages over methods from causal representation learning which aim to learn disentangled representations directly as part of the model architecture?
- What are the primary methodological advantages of using TopK SAEs for learning concepts from embeddings compared to alternative approaches for learning disentangled representations?
- What are the potential limitations of defining a counterfactual as the nearest neighbor that does not activate a concept, particularly the risk that confounding biological differences between the two cells might influence the gene attribution scores?
- Which measures were taken to assess intra-expert consistency?
- Can you provide an example where your model identifies a concept that reveals a previously non-obvious connection between genes or refines existing knowledge in a way that leads to a new, testable hypothesis?

---

> ### Author Response · Authors · 2025-11-21
>
> Dear reviewer, we sincerely appreciate your feedback regarding both the strengths and the areas for improvement in our paper, which will help us enhance the quality of our work. First of all, thank you for noticing the typos, we corrected them. Please find our answers to your questions below.
>
> **W1/Q1**: The need to clarify the positioning was also mentioned by other reviewers, please find our answer in the common answer (number 1).
>
> **W2/Q2**: The need to clarify the choice of the decomposition method was also mentioned by other reviewers, please find our answer in the common answer (number 4).
>
> **W3/Q3**: The motivation for using counterfactual perturbations instead of more standard ones (such as replacing gene expression with 0) is to keep the perturbations within a realistic neighborhood, thereby preventing large shifts that might capture broad biological differences unrelated to the target concept. We agree that our approach only reduces potential confounding biological signal, but we believe that the remaining risk is substantially mitigated by averaging over several counterfactual perturbations and for several prototypes.
>
> **W4/Q4**: Thank you for recognizing the value of the expert study. We acknowledge its limitations and conducted an additional experiment to assess intra-expert consistency, as you suggested. Please find our detailed answer in the common answer (number 2).
>
> **W5/Q5**: We agree that this is a claim and that it corresponds to a promising application of this framework in future work. We are currently exploring the potential for knowledge discovery and don't yet have proper results to show in this contribution. We mentioned it in the introduction and conclusion as an opening, do you think that we should soften the formulation?

---

> > ### Comment · Reviewer_YuVv · 2025-11-22
> >
> > Dear authors,
> >
> > Thank you for your thorough response to all the weaknesses and questions I raised. W/Q1–4 have been addressed to my satisfaction.
> >
> > Regarding W/Q5, I suggest softening the formulation.
> >
> > Given that the majority of my concerns have been successfully addressed, I will raise my rating accordingly.

---

> > > ### Author Response · Authors · 2025-12-01
> > >
> > > Dear reviewer,
> > >
> > > Thank you for your review, which helped us strengthen the manuscript. Following your suggestion, we have softened the formulation in the introduction. We are pleased that our answers and revisions successfully addressed your concerns.

---

### Official Review · Reviewer_GBNb · 2025-10-31

**Soundness:** 2
**Presentation:** 3
**Contribution:** 2
**Rating:** 2
**Confidence:** 4

**Summary:**

The authors propose a concept-based interpretability framework for single cell RNA-seq data to uncover genes that activate different (biological) concepts, enabling interpretation of known single-cell emeddings. They also provide an interactive interface to facilitate the concept analysis by experts, as well as an enrichment method based on uncovered attributes.

The authors show that the purpose model leads to better interpretability compared to individual neurons.

**Strengths:**

The paper provide a novel way of interpreting the embeddings provided by well known scRNA-seq models such as scVI and scGPT. They also provide an intuitive framework to analyze the obtained results by experts that may not be able to run the model themselves. I think this work can have important relevance on the biomedical community, as interpretability is a key factor while working with single-cell data.

**Weaknesses:**

I don't think this work provide enough sound advances in representation learning or machine learning for being suitable to ICLR. The method applies known SAE for dictionary learning based on known representation learning models from scRNA-seq data (scVI, scGPT). While the idea of interpreting the concepts inferred by the SAE is very interesting, they are more suitable to more specialized conferences/journals.

In addition to that, I believe the results are somehow incomplete. While they provide good analyses on the SAE across the two embedding spaces, a more in-depth benchmark is required to fully understand the benefits of the proposed model. For example, there are other integration/embedding models that are widely used that should be compared such as CCA (Seurat), Harmony, the new U-space by MrVI, etc. The inclusion of these models could provide a more "universal" view of the proposed interpretability framework.

Additionally, more there are other works based on SAE for interpretability (eg. https://doi.org/10.1093/nar/gkae197, and reference therein) that should be mentioned (and maybe compared).

Finally, the expert analysis (the soundness of the biology inferred by the model) should be more thoroughly analyzed (see for example, the work mentioned above).

**Questions:**

How does the model perform when using other integration/embedding models for scRNA-Seq data?

How does it compare against traditional GSEA analysis on the same comparisons? I understand that through GSEA genes themselves may not have "interpretability" but are both models uncovering the same biological processes?

---

> ### Author Response · Authors · 2025-11-21
>
> Dear reviewer, thank you for recognizing the relevance of our work and for your feedback on the weaknesses, which will help us improve the manuscript. Please find below our answers to your questions:
>
> **W1**: Thank you for acknowledging that our approach is interesting in the context of applications to biology, which is the track to which we submitted. We agree that the use of SAE is not novel, however, interpreting concepts from black-box models is still challenging for many data modalities, including single-cell RNAseq. We believe that we proposed novel methods to tackle this challenge, with counterfactual perturbations and attribution-based GSEA.
>
> **W2/Q1**: We agree that it would be interesting to apply the framework to other models; however, the objective of this work was not to benchmark several embedding models but rather to introduce methods to interpret the latent space of single-cell RNA-seq encoders. We decided to first focus on two well-known models and specifically chose models with different architectures (a VAE and a Transformer).
>
> **W3**: Thank you for mentioning these works. As required by other reviewers also, we will clarify the positioning regarding the interpretable-by-design approaches based on interpretable/disentangled latents. Please find a detailed answer in the common answers (number 1). We don't think that the comparison to these works is in the scope of the paper, as we focus on black-box models and post-hoc interpretation.
>
> **W4**: We agree that our interpretation framework can be further applied to analyze black-box encoder models more in-depth. This is a challenging task, and we believe that our approach provides the community with an additional tool to address it. In the paper, we present several analyses, both quantitative (pathway-enrichment analysis, user study) and qualitative (interpretation of multiple concepts throughout the manuscript, identification of concepts related to the cell cycle with a probing task, identification of stable concepts across datasets with a stability metric).
>
> **Q2**: Thank you for suggesting this analysis. We conducted an additional experiment, please find the answer in the common answer (number 3).

---

### Official Review · Reviewer_mM9A · 2025-10-31

**Soundness:** 3
**Presentation:** 3
**Contribution:** 3
**Rating:** 6
**Confidence:** 2

**Summary:**

This ms mainly focuses on exploring the cell embeddings given by scGPT and scVI. The idea is to perform a low-rank decomposition of the embedding matrix (cell x emb) using sparse autoencoder (SAE), yielding a loading matrix U and factor matrix D. The factor matrix is called "concept" by the authors. Then they try to see which genes are relevant for the concepts. This is done by masking each input gene on chosen cells and check how they change the loading of a given concept. The change is called "attribution" by the authors. In two experiments, they show that the concepts are biologically interpretable, which are associated with cell types, GO pathways. This work is a step further than traditional tasks of using cell embeddings for clustering or batch effect correction.

**Strengths:**

1. The idea of introducing concept seems to make biological sense.
2. I think the association between the concepts and GO terms (Fig. 3B and Fig. 4C) are very interesting.  It could be further explored in the future.
3. Although missing some details, the method descriptions are solid.

**Weaknesses:**

Maybe I have missed, but In section 4.2, it is unclear to me
1. the distribution of the attribution scores, why 0.05 is selected as the cutoff
2. what are the cells used for calculating the attribution scores

Lack of a baseline method for the benchmark. How does it compare with direct low rank decomposition of the cell embedding matrix? In theory, you can treat the factors as the "concepts" and loadings as the "concept activations".

**Questions:**

line 179, why not directly mask gene l in x^p ?
line 288, what is "expansion factor" and why it matters here?
line 300, what is "neurons" in scGPT? could you be more specific about the layers?

---

> ### Author Response · Authors · 2025-11-21
>
> Dear reviewer, thank you for your feedback on both the strengths and weaknesses of our contribution. Please find below our answers to your questions:
>
> **W1**: (1) We added the cumulative distributions of attribution scores in the Appendix (Figure 11). The threshold we chose is restrictive and only selects the most important genes; it is mainly used to compare SAEs with different hyperparameters. In practice, we can instead show N genes to the user (N being limited by how much information a user can process), and the attribution-based GSEA only requires the list of genes sorted by attribution scores. (2) The attribution scores are computed on $N_p=10$ prototype cells, which are the cells with the highest concept activation scores.
>
> **W2**: Thank you for your suggestion. We could indeed use direct low rank decomposition of the cell embedding matrix. These approaches were grouped with SAEs under a unified framework in [1]. The objective of our work was not to benchmark several concept extraction approaches, but rather to introduce concept interpretation methods that are applicable to any concept extraction method. As this point was mentioned by other reviewers, please find a more detailed answer in the common answer (number 4).
>
> [1] Fel, Thomas, et al. "Archetypal SAE: Adaptive and Stable Dictionary Learning for Concept Extraction in Large Vision Models." Forty-second International Conference on Machine Learning.
>
> **Q1**: In the case of scVI, the model requires the full sequence of input genes, so masking a gene would mean replacing its expression by a masking value such as 0. We argue that 0-perturbations are not appropriate in our setting because: (1) they correspond to large perturbations that can create biologically irrelevant cells, causing the method to detect biological signals unrelated to the concept, and (2) they fail to capture situations where a concept detects under-expression of a gene.
>
> **Q2**: The expansion factor corresponds to the ratio $\frac{\text{number of concepts}}{\text{number of neurons}}$. In the literature, it is usually set very large for LLMs (more than 20 000 in [1]), but smaller for models in biology (3 in [2]). It directly controls the number of concepts $c$, a key hyperparameter, while being comparable between models of different sizes.
>
> [1] Gao, Leo, et al. "Scaling and evaluating sparse autoencoders." The Thirteenth International Conference on Learning Representations.
>
> [2] Adams, Etowah, et al. "From Mechanistic Interpretability to Mechanistic Biology: Training, Evaluating, and Interpreting Sparse Autoencoders on Protein Language Models." Forty-second International Conference on Machine Learning.
>
> **Q3**: The "neurons" refer to the cell embeddings that we decomposed with the SAE. For scGPT, it is the "CLS" token of the last layer. Thank you for highlighting that this implementation detail was unclear; we have clarified it in the revised version.

---

> > ### Comment · Reviewer_mM9A · 2025-11-26
> >
> > I think it is necessary to perform benchmark with direct low rank decomposition methods. While authors argue that this has been done in previous papers, it is not done specifically to the scRNA-seq data here. As this is an application paper, it is necessary to make it clear where what is SOTA and what is the added value of the proposed method. If low rank decomposition method can find the same "concepts", then the added value the proposed approach is not sufficient. After reads the responses from authors and comments from other reviewers, I will lower my score to this paper.

---

> > > ### Author Response · Authors · 2025-11-28
> > >
> > > Dear reviewer,
> > >
> > > Following your recommendation, we conducted additional experiments on scRNA-seq data to compare TopK SAE with a classic decomposition method. Please find the complete results in Appendix B (Tables 4 and 5).
> > >
> > > Following [1] and because the model activations are not non-negative, we used the semi-NMF decomposition method and found consistent results: for the same number of concepts $c$ and comparable sparsity levels, the decomposition approach has weaker reconstruction performance: R² of 0.980 (TopK SAE) vs. 0.849 (semi-NMF) for scVI, and 0.995 (topK SAE) vs. 0.933 (semi-NMF) for scGPT.
> > >
> > > We also evaluated the preservation of biological signal, using the cell cycle phase and cell type classification tasks described in Section 4.5 and Appendix G. While TopK SAE concepts closely match the performance of neurons, the accuracy decreases with semi-NMF concepts : for cell type classification, accuracy of  0.85 (topK SAE) vs. 0.79 (semi-NMF) for scVI, and 0.86 (topK SAE) vs. 0.73 (semi-NMF) for scGPT, indicating a loss of biological signal.
> > >
> > > We also evaluated the low-rank setting with $c < d$ and found a marked decrease in reconstruction performance and downstream tasks accuracy.
> > >
> > > In addition, the decomposition method is less practical at inference time, as it requires multiple optimization steps rather than a single forward pass, as in TopK SAE.
> > >
> > > We added the results in Appendix B. Thank you for suggesting this experiment, which further justifies the use of TopK SAE in the context of single-cell RNA-seq data. We believe it indeed leads to a strong addition to our paper value.
> > >
> > >
> > > [1] Fel, Thomas, et al. "Archetypal SAE: Adaptive and Stable Dictionary Learning for Concept Extraction in Large Vision Models." Forty-second International Conference on Machine Learning.

---

### Author Response · Authors · 2025-11-21
**Common answer (1/3)**

Dear reviewers, we sincerely appreciate the time and effort spent reviewing our manuscript and thank you for your detailed feedback. Because some questions or limitations were raised by multiple reviewers, we have prepared a summary of our main answers, additional experiments, and the corresponding revisions made to the paper:

**1. Clarify the positioning and differences with disentangled representation learning / interpretable latent spaces by design (reviewers YuVv, Tc45, and GBNb)**

In contrast to approaches that enforce interpretability of the latent space at training time (NetActivity, discrepancy-VAE, SENA, GEARS, VEGA, GSAE), our approach is post-hoc and model-agnostic, enabling interpretation of any trained neural network. This distinction offers several advantages: (1) it allows interpretation of black-box models that may achieve superior predictive performance compared to interpretable-by-design architectures, (2) it does not require prior knowledge to be specified at training time, facilitating the incorporation of new biological insights as they emerge, and (3) it could enable the discovery of novel biological patterns.

Modifications to the paper: We clarified this positioning in the "Related work" section and in the introduction.


**2. Limitations of the user study (reviewers YuVv, Tc45, and GBNb)**

The reviewers pointed out the limitations of the user study, which, although valuable, involves only a single participant. We acknowledge this limitation, which mainly reflects cost and time constraints, as the study requires experts in biology. As suggested, we conducted an additional analysis to assess intra-user variability. We presented 14 concepts to the domain expert, 11 of which had been annotated in the original user study. Among these 11 concepts, 5 received similar interpretations, 4 switched between "Signal but unclear" and "Not interpretable" (two in each direction), 2 switched between "Not interpretable" and "Interpretable" (one in each direction). These observations illustrate the sensitivity of the "Signal but unclear" annotation. We also observe variability even in annotations we expected stronger, which is more concerning, though difficult to characterize with such a small number of re-annotated concepts. Re-evaluating all concepts a second time would undoubtedly bring deeper insights. While this cannot realistically be done within the limited rebuttal timeframe, it remains an important step that we will address in the next steps of the work.

Modifications to the paper: we included a new paragraph in Section 4.3 discussing the limitations of the user study based on the results of the intra-user stability experiment.

---

### Author Response · Authors · 2025-11-21
**Common answer 2/3**

**3. Compare attribution-based GSEA to classic DGEA-based GSEA (reviewers Tc45 and GBNb)**

Attribution-based GSEA is a methodological contribution of our work, the objective is to identify biological processes that best match the signal encoded by the concept. We showed with deletion curves (Figure 8 in Appendix) that genes with high attribution scores have more impact on the concept activation than genes with high fold change obtained from classic differential gene expression analysis (DGEA). As suggested by reviewers, comparing attribution-based GSEA to classic DGEA-based GSEA would further justify the advantages of our method.

Additional experiment: We computed DGEA-based GSEA and attribution-based GSEA of 100 concepts for the two models. For a given concept, we obtain two sets of biological processes :

- Biological processes from attribution-based GSEA : $P_{att} = {T^{att}_1, ... ,T^{att}_{k_{att}}}$
- Biological processes from DGEA-based GSEA : $P_{dgea} = {T^{dgea}_1, ..., T^{dgea}_{k_{dgea}}}$

Each biological process $T$ is defined as a set of genes $T={(g_1, l2fc_1, att_1), ..., (g_m, l2fc_m, att_m)}$, with $g$ the gene name, $l2fc$ the log2 fold change from DGEA and $att$ the attribution score.

We compared the results only if there is at least one biological process in both $P_{att}$ and $P_{dgea}$. The metrics we compute are :
- The maximal absolute l2fc value in the biological processes: $\max_{T \in P_{\mathrm{att}}}  \max_{(g, \mathrm{l2fc}, \mathrm{att}) \in T} |\mathrm{l2fc}|$
- The maximal attribution score in the biological processes: $\max_{T \in P_{\mathrm{att}}} \max_{(g, \mathrm{l2fc}, \mathrm{att}) \in T} \mathrm{att}$
- IoU of pathways : $\frac{\left| P_{\mathrm{att}} \cap P_{\mathrm{dgea}} \right|}{\left| P_{\mathrm{att}} \cup P_{\mathrm{dgea}} \right|}$
- IoU of genes : $\frac{\left| G_{\mathrm{att}} \cap G_{\mathrm{dgea}} \right|}{\left| G_{\mathrm{att}} \cup G_{\mathrm{dgea}} \right|}$ with $G_{\star} = \bigcup_{T \in P_{\star}} \{ g \mid (g, \mathrm{l2fc}, \mathrm{att}) \in T \}$

Results: First, we observe minimal overlap between the biological processes detected by the two methods (mean IoU of 0.068 for scGPT and 0.025 for scVI) and between the genes within these processes (mean IoU of 0.055 for scGPT, 0.027 for scVI), demonstrating the need to choose one of the methods. As expected, biological processes identified through classic DGEA-based GSEA contain genes with higher absolute log2 fold change (mean 4.0 vs. 2.6 for scGPT, 4.7 vs. 2.4 for scVI), whereas the biological processes identified with the attribution-based method contain genes with higher attribution scores (mean 0.32 vs. 0.17 for scGPT, 0.21 vs. 0.14 for scVI). Deletion curves (shown in the Appendix) further indicate that genes identified via attribution exert a greater impact on concept activation than those identified via DGEA. Together, these results justify the use of attribution-based GSEA for concept interpretation, as the resulting biological processes more accurately reflect the signal associated with the concept.

Modifications to the paper: we have added this analysis to Appendix C.

---

### Author Response · Authors · 2025-11-21
**Common answer (3/3)**

**4. Further justify the choice of the decomposition approach (reviewers YuVv and mM9A)**

The reviewers requested justification for the use of TopK SAE over other decomposition approaches, such as different SAE architectures or low-rank decomposition methods.
The objective of this work is not to compare several decomposition approaches, but rather to introduce methods for interpreting and validating the extracted concepts. The methods and framework we propose are not specific to TopK SAEs and other concept extraction methods could be used seamlessly, such as Non-Negative Matrix Factorization (NMF, used in [1] for instance) or other SAE architectures like vanilla SAE [2] or Matryoshka SAE [3]. We chose SAEs over decomposition methods like ICA or NMF following the results of [4], which shows that SAEs have lower reconstruction errors at fixed sparsity. We then decided to use TopK SAE as it simplifies tuning and improves the reconstruction-sparsity frontier compared to vanilla SAE [5].

Modifications to the paper: we clarified the choice of TopK SAE in the "Background on concept extraction" section.

[1] Fel, Thomas, et al. "A holistic approach to unifying automatic concept extraction and concept importance estimation." Advances in Neural Information Processing Systems 36 (2023): 54805-54818.

[2] Huben, Robert, et al. "Sparse autoencoders find highly interpretable features in language models." The Twelfth International Conference on Learning Representations. 2023.

[3] Bussmann, Bart, et al. "Learning Multi-Level Features with Matryoshka Sparse Autoencoders." Forty-second International Conference on Machine Learning.

[4] Fel, Thomas, et al. "Archetypal SAE: Adaptive and Stable Dictionary Learning for Concept Extraction in Large Vision Models." Forty-second International Conference on Machine Learning.

[5] Gao, Leo, et al. "Scaling and evaluating sparse autoencoders." The Thirteenth International Conference on Learning Representations.

---

### Author Response · Authors · 2025-12-03

Dear Area Chairs,

In light of the specific review conditions, we would like to provide a summary of the improvements made to the manuscript in response to the reviews. We are pleased that our manuscript was recognized as clearly written and well-structured, with solid methodological descriptions (reviewers Tc45, YuVv, mM9A), and that our contribution on the interpretability of large scRNA-seq foundation models is considered highly relevant (reviewers Tc45, GBNb).

We believe we have successfully addressed most of the concerns raised in reviews (as stated by reviewers YuVv and Tc45) and are grateful to the reviewers for their constructive suggestions, which helped us improve the manuscript. In particular, we conducted additional experiments :
- **Decomposition method for concept extraction**: We clarified the choice of TopK SAE in the “Background on concept extraction” section with supporting literature and conducted an additional experiment reproducing results on single-cell RNA-seq data (Appendix B), which further validates this choice.
- **Comparison of our attribution method with classic DGEA**: Reviewers requested further justification for the methods we introduce to interpret concepts. We conducted an additional experiment comparing our attribution-based GSEA with classic DGEA-based GSEA and added the results to Appendix C. The experiment further demonstrates how our method enhances concept interpretation, a challenging step in unsupervised concept-based approaches, particularly for complex modalities.
- **User study**: We conducted an experiment assessing intra-expert reproducibility and added the results to Section 4.3. While some limitations remain due to time constraints, we believe the study provides valuable insights, as such evaluations are rarely included in similar works.

We sincerely appreciate your time and consideration.

Best regards

---

### Meta-Review · Area_Chair_oaVe · 2026-01-07

**Summary:**

In general, reviewers agree on the potential merits of the proposed concept-based interpretability framework for scRNA-seq foundation models and the importance of enhancing interpretability in scRNA-seq analysis.
Reviewers also note several limitations of the current study, which includes limited methodological innovation, insufficient evaluation to convincingly demonstrate the advantages of the proposed framework, limited expert validation, need for providing a clearer context of the proposed interpretability framework based on relevant prior work in the field, and concerns regarding the counterfactual analysis and claims regarding the ability for hypothesis generation.

**Reviewer Concerns:**

The authors have responded to the respective concerns raised by the reviewer.
The authors rebuttal clarifies by positioning and significance of the proposed framework, presents additional study on intra-expert reproducibility, further justification and comparison for approaches taken by the authors for concept interpretation.
Overall, the authors responses clarifies several minor doubts and enhances the clarity of the presentation.
However, I do not think they provide strong arguments and further evidence to substantially change the reviewers' original rating.

**Reviewer Scores:**

Reviewer mM9A, who was originally relatively positive (evaluation score of 6) regarding the manuscript expressed concerns regarding the lack of additional evaluations to clearly understand the improvement and benefits of the proposed approach.
This has been later addressed by the authors to some extent, but it is unlikely that the reviewer might have increased the score beyond 6 based on the discussion between the reviewer and the authors.

Reviewer GBNb has substantial concerns regarding the original manuscript, giving a low rating of 2 initially.
However, the authors provide only brief clarifications in response, and I doubt that might have significantly changed the reviewer's original position, as the rebuttal does not provide strong arguments against the reviewer's concerns.

Reviewer YuVv has mostly requested for further clarification & justification, and it is likely that the authors responses may have addressed them to a certain extent.
I expect that the original rating of 4 may have slightly increase to 5 or so.

Finally, reviewer Tc45 had concerns about novelty, limited expert validation, mostly qualitative assessment of the interpretability gain, and limited evaluation data.
While the authors have provided brief response to most of the concerns, the reviewer still has outstanding concerns about limited novelty and expert validation, and it is highly likely that the original score might have been mantained.

---

### Decision · Program_Chairs · 2026-01-26

Reject